# A Multi-Site Observational Evaluation of the Equine Assisted Growth and Learning Association Model of Equine-Assisted Psychotherapy for Veteran Trauma Survivors

**DOI:** 10.3390/ijerph22101557

**Published:** 2025-10-13

**Authors:** Halina Kowalski, Hannah Van Buiten, Patricia Hopkins, Connie Baldwin, Elena Nazarenko, William R. Marchand

**Affiliations:** 1Independent Researcher, Bend, OR 97702, USA; 2Independent Researcher, Kailua, HI 96734, USA; 3Psychology Department, Augsburg University, Minneapolis, MN 55454, USA; 4Equine Assisted Growth and Learning Association, Woodinville, WA 98072, USA; connie.baldwin@eagala.org; 5Whole Health Service, VA Salt Lake City Health Care System, Salt Lake City, UT 84148, USA; 6Department of Psychiatry, University of Utah, Salt Lake City, UT 84108, USA; 7Animal, Dairy and Veterinary Sciences, Utah State University, Logan, UT 84322, USA

**Keywords:** military, veterans, Equine Assisted Growth and Learning Association, Eagala, psychotherapy incorporating horses, trauma, complementary and integrative interventions, PTSD, equine-assisted psychotherapy, equine-assisted services

## Abstract

The primary aim of this study was to evaluate the feasibility, safety, and preliminary outcomes of the Equine Assisted Growth and Learning Association (Eagala) model of equine-assisted psychotherapy for active-duty military and veteran trauma survivors. This was a retrospective multi-site observational study. Study participants completed four psychological instruments pre- and post-intervention. These were the PTSD Checklist for DSM-5, the Patient Health Questionnaire, the Satisfaction with Life Scale and the Sheehan Disability Scale. The Client Satisfaction Questionnaire-8 was also completed post-intervention. Paired-sample *t*-tests were conducted to assess for changes in the primary outcome variables pre- and post-intervention. The RAPID qualitative approach was used to analyze the qualitative data and develop subthemes. Subjects were 107 participants at 12 sites. Participants ranged in age from 22 to 78 and were predominately male. Findings revealed that the Eagala model intervention can be implemented for this population across multiple sites. Further, treatment engagement may be better than found with conventional psychotherapy interventions for this population. Pre-to-post-intervention changes in scores on the psychological instruments revealed significant decreases in PTSD symptoms, depression, and disability as well as increases in satisfaction with life. Future randomized controlled trails of this intervention are warranted.

## 1. Introduction

Natural disasters, wars, and interpersonal assaults lead to worldwide trauma exposure [1], and as a result, over 3.5% of the world’s population may experience posttraumatic stress disorder [2]. Among US military veterans, one study [3] found that 87% had been exposed to at least one potentially traumatic event. Thus, rates of posttraumatic stress disorder (PTSD) among military personnel and veterans are around 30% [4,5] compared to the lifetime prevalence of PTSD among adults in the US general population, which is estimated to be 6.8% [2,6]. Those with PTSD often experience impairing symptoms; reduced interpersonal, social, and occupational functioning; a lowered quality of life; and physical health problems, as well as an elevated suicide risk [6]. In addition to PTSD, trauma exposure can result in the development of other psychiatric disorders [7] such as major depressive disorder [2] and functional impairment [7]. Thus, interventions are needed for veteran trauma survivors with and without a diagnosis of PTSD.

Conventional pharmacological and psychological treatments are available for the subgroup of trauma survivors with PTSD. Eye Movement Desensitization and Reprocessing, Cognitive Processing Therapy, and Prolonged Exposure are often considered gold standard therapies [8]. However, as many as one-half of veterans receiving these interventions do not experience significant improvement [9]. In addition to an inadequate treatment response, many veterans do not engage in, or complete, conventional psychotherapeutic interventions [4,10,11,12]. Lastly, conventional interventions for PTSD may not fully address other sequelae of trauma [13]. These include effects of military sexual trauma with rates as high as 15% among female veterans [11], trauma-related guilt [14], disrupted attachment [15], and moral injury [16]. Thus, there is a need to investigate interventions for trauma survivors that may enhance treatment engagement and/or outcomes associated with conventional treatments [17,18] and/or might be equally, or more, effective as monotherapy than those currently available.

In response to the above, the utilization of complementary interventions is being increasingly explored for veteran trauma survivors. Among complementary interventions with potential to help veteran trauma survivors, animal-assisted interventions (AAIs) are a category of therapies that use animals to help humans [19]. AAIs are increasingly utilized as complementary interventions for trauma survivors in general [20,21,22] and veterans [14,15,23,24,25,26,27,28,29,30,31,32,33,34,35,36,37,38,39,40,41]. AAIs that utilize horses fall under the umbrella term of equine-assisted services (EASs) and include interventions such as therapeutic riding, equine-assisted learning (EAL), and equine-assisted psychotherapy (EAP) [19]. There is evidence that EASs may benefit both veteran [14,15,23,24,25,26,27,28,29,30,31,32,33,34,35,36,37,38,39,40,41,42,43,44,45,46] and non-veteran [47,48,49,50,51] trauma survivors. However, rigorous studies are mostly lacking [13], and the field is under-resourced and thus underdeveloped. For example, among twenty-three published studies on veteran populations [14,15,23,24,25,26,27,28,29,30,31,32,33,34,35,36,37,38,39,40,41,42,43,44,45], only three [24,29,30] had a control group and only one [29] was a randomized trial. Nonetheless, EAS, including EAP, interventions are being increasingly used for community populations [52], as well as for military service members and veterans with trauma histories [23]. Thus, rigorous research is needed to move the field forward.

A challenge to the field is that many different EAP interventions are currently being utilized for veterans with trauma histories, and therefore, interpreting and drawing conclusions from the existing literature is difficult [13]. To move the field of EAP for veterans forward, the investigation of structured interventions that facilitate both fidelity to the model in replication studies and multi-site randomized controlled trails will be needed [13]. A promising development is in the published reports of structured interventions that could be manualized [14,23,30,34,35,40] and therefore facilitate both large replication studies and, ultimately, if shown to be effective, dissemination to the field.

The primary aim of this study was to move the field of utilizing EAP for veterans with trauma histories forward by evaluating the feasibility, safety, and preliminary outcomes of implementing a well-established intervention that utilizes a structured therapeutic framework across multiple sites. Specifically, we investigated the use of the Equine Assisted Growth and Learning Association (Eagala) (https://www.eagala.org/index, accessed on 18 January 2025) model of EAP among active-duty military and veterans with trauma histories across multiple sites.

The Eagala model provides a structured therapeutic framework that balances consistency across treatment teams with flexibility for client-led exploration. Central to the model is the integration of horses as co-facilitators; their genuine presence and responsiveness shape the therapeutic process through ground-based non-riding sessions that foster mutual respect and natural interactions. Each treatment team includes a licensed or registered mental health professional (MH), a qualified equine specialist (ES), and at least one horse, but most often a herd, working together to deliver solution-focused sessions tailored to client needs [53,54,55,56].

Horses, like humans, are social animals with defined herd roles, unique personalities, and emotions that mirror human relationships, making them natural partners in metaphorical learning. Building trust with them requires resilience and persistence, engaging people physically, emotionally, and mentally in ways that counter today’s culture of instant gratification. Their large, powerful presence can feel both intimidating and inspiring, inviting clients to build confidence alongside them while practicing healthy relationship skills. As prey animals, horses remain deeply focused on safety and survival, responding with flight, fight, or freeze responses while staying highly attuned to non-verbal cues. In therapeutic settings, this sensitivity allows horses to reflect clients’ relational dynamics, whether with family, colleagues, or inner struggles, so that horses emerge as living symbols and co-facilitators, offering a safe opportunity to explore, practice, and transform patterns together [56].

The mental health professional’s role is the same as with other therapies: they will be responsible for documentation, working with the ES to ensure sessions are set up to align with the client’s treatment goals, maintaining the emotional safety for all members in the session, and following the Eagala framework during discussions. The equine specialist’s role is to oversee all horse management, help with choosing horses and activities, work with the MH during each session to provide horse observations, and ensure physical safety. Both members work together during the session, staying close to one another and making decisions based on the presentation of horses and clients in real time. The team will often move into the background of the session, allowing the horse-and-client dynamic to take shape [56].

The Eagala model is grounded in the belief that clients can discover their own best solutions through experiential learning [56]. Facilitators must complete in-person training, and demonstrate competency through practical and online examinations, and may pursue the Military Service Designation for work with service members and veterans (https://www.eagala.org/index, accessed on 18 January 2025). The model is guided by professional standards emphasizing a team approach, ground-based experiences, a solution-oriented framework, and adherence to a code of ethics, and can be delivered in individual, group, or couples’ formats.

The Eagala model in practice can be organized into two components: session design and facilitation skillsets. Session design is structured around four categories, each offering different ways in which clients and horses may engage with one another: Move, Relate, Observe, and Create. Move invites clients to notice how their actions influence horse movement or stillness, symbolizing change, grounding, or feeling stuck. Relate positions the horse as a relational partner, encouraging reflection on patterns of connection. Observe focuses on watching horses’ natural behaviors without direct interaction, fostering perspective taking and awareness. Create involves designing symbolic representations of treatment goals or challenges and observing how horses engage with them. These categories provide structure while allowing individualized outcomes [56].

The facilitation skillsets in the Eagala model include the SPUd’S framework (which categorizes equine responses into Shifts, Patterns, Unique behaviors, and Discrepancies in relation to client reactions), the use of non-interpretive language, intentional pauses in verbal interventions, and the encouragement of metaphor and projection. In practice, the treatment team uses the SPUd’S framework to take note and reflect horse behaviors to clients by using non-interpretive language, drawing on the client’s own words to frame questions that invite metaphor development, projection, and experiential learning. By observing horse responses, such as movement, patterns, and unique behaviors, as potential symbolic representations, the SPUd’S framework provides a neutral anchor for discussion and supports a structured process of observation, projection, metaphor building, and integration. Another component of the Eagala framework is the use of metaphors, which are things representative or symbolic of something else, to help clients externalize struggles, reduce emotional intensity, and create space for insight. While working with the horses, clients may also engage in projection, which is unconsciously attributing their own feelings or conflicts onto others. Used together, metaphor and projection transform internal experiences into symbolic form, providing distance from overwhelming emotions and opening pathways to new perspectives and self-understanding. Equally central to the Eagala framework is the use of intentional pausing of verbal interventions. This gives clients time to regulate affect, process material, and reflect before responding. Silence reduces facilitator over-direction, reinforces the agency of the client, and functions as a coregulation strategy. Within experiential models like Eagala, these pauses allow clients to notice and integrate metaphoric and relational dynamics, securing insight in lived experience and reinforcing self-directed discovery and growth.

The Eagala model provides a structured therapeutic framework as outlined above but does not dictate the specific activities of sessions. Upon arrival for a session, clients identify the focus of their work for the day, and the treatment team designs an activity within the session framework of Move, Relate, Observe, or Create. This client-driven process establishes a dynamic flow grounded in real-time needs and emphasizes in-the-moment interactions with the horses rather than provider-directed activities. Given that the horses serve as metaphors, no specific instructions are provided regarding how clients should interact with horses.

Considered as a whole, the Eagala model framework integrates structured session design and facilitation, professional standards, and the authentic presence of horses to foster psychological insight, experiential learning, and opportunities for clients to practice new behaviors in a safe, low-risk environment supported by the nonjudgmental partnership of horses [57].

Principal findings revealed that the Eagala model of EAP can be successfully implemented for active-duty military and veteran trauma survivors across multiple sites in the US. Also, treatment engagement may be better than that found with conventional psychotherapy interventions for this population. Finally, preliminary findings suggest that improvements in PTSD and depressive symptoms, as well as decreased disability and enhanced satisfaction with life, may be associated with participation. Future randomized controlled trails of this intervention are warranted.

## 2. Materials and Methods

This retrospective observational study reported the analyses of quality improvement (QI) data collected from 12 Eagala military-designated therapeutic equine facilities in the US that received funding support from the Eagala organization and provided services to active-duty military and veteran service members from 1 September 2018 until 30 September 2019. Eagala model EAP was provided as clinical programming by the various facilities, not for research purposes. However, the facilities did collect quality improvement (QI) data on their programming, which was then collected retrospectively from the facilities by the Eagala organization for this study. The sample reported herein represents the participants for which complete data sets were obtained.

Active-duty and military veterans with trauma histories who enrolled to participate in Eagala-model clinical programming were invited to participate in the QI data collection at the various sites. Participants were informed that participation in the data collection was optional and that the Eagala organization intended to utilize the deidentified outcomes of these QI measures in aggregate for reports and publications. Those that chose to do so consented and completed the psychological assessments described in Section 2. Those that did not wish to participate in the study were allowed to participate in the full clinical program without penalty and did not have their information represented in the research sample. The only inclusion criteria for data collection were active-duty or military veteran status and having a history of trauma and self-identifying as having a disability. The exclusion criteria were that the veteran or military member could not have a dishonorable discharge or been barred from receiving Veterans Administration pension, services, or other benefits. Data regarding the number of program participants who chose to participate in the QI project was not collected.

Participants who consented (IRB number 00177310) to participate in data collection completed an assessment packet prior to the first session, which included locally developed questionnaires regarding demographic information, military service history, and mental health history. Additionally, study participants completed four psychological instruments immediately pre- and post-intervention, administered by the mental health clinician facilitating the intervention. These were the Standard Civilian PTSD Checklist for DSM-5 [58], the Patient Health Questionnaire [59], the Satisfaction with Life Scale [60], and the Sheehan Disability Scale [61]. Lastly, Client Satisfaction Questionnaire-8 [62] was completed post-intervention.

The Standard Civilian PTSD Checklist for DSM-5 (PCL-5) is a 20-item scale that evaluates the severity of PTSD symptoms [58]. The questions are in a 5-point Likert scale format that ranges from 0 (‘Not at all’) to 4 (‘Extremely’), and the total scores range from 0 to 80 with higher scores corresponding to greater PTSD symptoms. The Patient Health Questionnaire (PHQ-9) is a 9-item scale that measures the severity of depressive symptoms; the higher the total score is, the more severe the symptoms of depression are [59]. Each of the nine questions are in a 4-point Likert scale format that ranges from 0 (‘Not at all’) to 3 (‘Nearly every day’). PHQ-9 scores can be used to assess depression symptom severity that range from mild (five and below), moderate (10–14), and moderately severe (15–19) to severe (greater than 20). The Satisfaction with Life Scale (SWLS) is a validated 5-item scale [60] with response options that range from 7 (‘Strongly agree’) to 1 (‘Strongly disagree’). Higher SWLS score indicate higher satisfaction with life, and SWLS positively correlates with other well-being measures. The Sheehan Disability Scale (SDS) is a 3-item scale that indicates functional impairment related to three domains: work/school, social life, and family life/home responsibilities [61]. Each domain is determined by one 10-point Likert scale item, which ranges from 0 (‘Not at all’) to 10 (‘Extremely’). The total score ranges from 0 to 30, with higher scores indicating greater impairment. Lastly, Client Satisfaction Questionnaire-8 (CSQ-8) was used to evaluate participant satisfaction with the intervention. CSQ-8 [62] is an eight-question structured survey, and items are scored on a Likert scale from 1 (low satisfaction) to 4 (high satisfaction). Total scores range from 8 to 32, with higher scores indicating greater satisfaction. CSQ-8 also includes three open-ended questions that ask participants to report what was most helpful and least helpful about the treatment. Participants were also prompted to provide suggestions to improve the program. The three questions were as follows: (1) What parts of the program were most helpful to you? (2) What parts of the program were least helpful to you? (3) What suggestions do you have for how the program can be improved?

Data were inspected to ensure that the variables met assumptions for parametric tests. CSQ-8 was skewed and kurtotic. To combat this issue, bootstrapping was used with 5000 samples for all CSQ-8 analyses. Paired-sample *t*-tests were conducted to assess for changes in the primary outcome variables pre- and post-intervention. To assess quantitative outcomes, independent-sample *t*-tests were conducted to determine differences between pre- and post-intervention results by treatment modality (individual or group). Cohen’s *d* [63] was used to assess effect sizes of small (<0.2), medium (0.5), and large (0.8).

Lastly, the CSQ-8 open-ended question results were analyzed. Next, the RAPID qualitative approach [64] was used to analyze the qualitative data and develop subthemes. The qualitative analysis was divided among four members of the research team, who independently picked out the themes and analyzed the responses for a subgroup of the sample. Each of the questions served as a theme, and the responses were summarized for each of the participants. The coders met together and discussed the subthemes, discussed any difficulties in coding, and came up with a consensus. These responses were reviewed by a person trained in qualitative methods and used to summarize the results.

## 3. Results

### 3.1. The Intervention

Across the 12 sites, the Eagala model intervention, described in Section 2, was provided as a series of eight group (*n* = 54 participants, 50%), or six individual (*n* = 52 participants, 49%), or couple (*n* = 1 participant, 1%) sessions. Group interventions had a range from three to seven participants (M = 5.65, SD = 1.18). All sessions were 60 min in length. Each session utilized at least one equine, but most had more. Information about the actual number of horses used in each session was not collected. The average number of sessions attended was 5.96 out of six individual (98%) or 7.5 out of eight group (98%) sessions possible, and across all modalities, the number of sessions attended ranged from four to eight. The Eagala model does not specify the activities of sessions. As stated in the introduction, clients identify the focus of their work for each session, and the treatment team designs an activity within the session framework of Move, Relate, Observe, or Create. Thus, session activities were variable and data regarding the specific activities that occurred during each session was not collected.

### 3.2. Participants

Subjects were 107 active-duty or military veteran service members. Participants ranged in age from 22 to 78 (M = 46.23, SD = 12.63) and were predominately male (*n* = 70, 65.4%). Table 1 provides additional demographic and diagnostic data.

### 3.3. Quantitative Results

The results of the independent-sample *t*-tests comparing pre- and post-intervention changes in scores on the four psychological instruments (Table 2) revealed decreases in PTSD symptoms (PCL-5), depression (PHQ-9), and disability (SDS) and increases in satisfaction with life (SWLS) with mostly medium-to-large effect sizes for the entire sample (*n* = 107). Next, pre- and post-intervention scores by treatment modality (individual treatment vs. group treatment) were assessed for 106 participants who engaged in either modality (one individual received couple treatment and was therefore not included). The results mirrored the previous findings, showing reductions in PTSD, depression, and disability symptoms with increases in satisfaction in life for both modalities with medium-to-large effect sizes (see Table 3 and Table 4).

Next, independent-sample *t*-tests were conducted to determine whether treatment modality (individual vs. group) had a differential impact on post-intervention outcomes. The results showed no difference between the modality with PTSD symptoms or treatment satisfaction (CSQ-8). However, participants who completed individual sessions reported significantly greater self-identified reductions in symptoms of disability and depression, as well as reporting greater satisfaction with life at the end of treatment compared to participants in the group condition (see Table 5). Importantly, there were no differences between the conditions in any of these measures at pre-intervention. Lastly, we investigated whether there was a difference in the sessions completed depending on treatment modality. Participants who had received treatment in groups attended significantly more sessions than participants who had received individual treatment (see Table 5).

### 3.4. SQ-8 Survey and Qualitative Analysis Results

Responses to the three open-ended questions from CSQ-8 were analyzed [63] using the RAPID Qualitative Analysis procedure [65] and prominent subthemes were determined. Tabulated responses by subtheme are presented in Table 6. Prominent subthemes are further developed in the following text and outlined in Table 7.

### 3.5. Qualitative Results—Most Helpful Parts of the Program

Participants in the study highlighted the profound impact of interaction with horses and the application of these activities to personal problems through symbolism and metaphors. Some of the respondents mentioned the calmness and ease they felt connecting to the horses. “Making correlations between horses’ actions and those of people/myself and drawing unexpected conclusions.”

Therapist and peer group interaction was another frequently mentioned benefit. Respondents mentioned that talking with counselors and peers had been beneficial in processing their emotions and building relationships. Veterans commented on improvements in processing emotions/thoughts, relationship building/trust, and gaining insight/awareness. Many described how the program activities had encouraged them to evaluate their own actions and thought patterns and motivated change. One highlighted their process of “Realizing how largely I was a part of the problems and participating in the healing process.”

Some of the participants mentioned that feeling mindful and/or present through activities such as breathing techniques was also helpful. Many of the participants mentioned multiple benefits; one participant wrote “all of it”, in response to what they had found most helpful, while others did not specify the most beneficial part of the program, and one participant simply wrote “change.”

### 3.6. Qualitative Results—Least Helpful Parts of the Program

There were also many participant responses mentioned that they were entirely satisfied or had no criticism listed. Not having enough time or sessions was a prominent concern. Many felt that the brevity of the program had limited their ability to fully benefit from the therapeutic process.

Although infrequent, some participants found that some of the activities were challenging or unhelpful, such as “picking out tools”, “lunch”, and “standing around…” Some of the participants mentioned feeling unsure about the purpose of some of the specific exercises or struggling to derive value from them. Access concerns, such as the long driving distance to get to the program location and the rush hour traffic, were some of the mentioned challenges.

### 3.7. Qualitative Results—Suggestions for Program Improvement

A common recommendation was to have more and/or more frequent sessions, indicating that some participants felt the current timeframe was insufficient for achieving their therapeutic goals. Many suggested that longer sessions or a more extended program would allow for deeper engagement with the activities and better outcomes.

Additionally, some participants recommended engaging more veteran-to-veteran interaction in the program. This feedback emphasizes the unique value that shared experiences among veterans can bring to therapeutic settings and mutual understanding. Increasing the program promotion was also discussed. One suggestion was to “reach out to university student veteran population.”

Some of the participants suggested increased expectation setting/instruction, including discussions of program expectations and reasons for the therapeutic activities, discussions of the theory behind the human relationship with the horses, and discourse regarding stress. A frequent recommendation was to engage in different activities such as horse riding. Another suggestion was “…going on walks around the property with the horses would have been helpful and therapeutic at times…”

The desire for wheelchair accessibility was also mentioned. Some participants wanted more consistency in the group program structure. “One of the weeks, a new person stepped in (new to me), which is fine… For me, it became a small distraction that I feel would have been more effective had the person been there from the start.” Other veterans want more flexibility, such as an option for a short overnight program retreat and interactions with different horses. Some of the participants were fully satisfied with the program, while others had no suggestions: “No suggestions. It flowed beautifully.”

## 4. Discussion

The primary aim of this study was to evaluate feasibility, treatment engagement, preliminary outcomes, and other characteristics associated with utilizing the Eagala model of EAP for military members or veterans with trauma histories. To our knowledge, this was both the first multi-site as well as the largest sample size investigation in the field of EASs for veterans reported to date. As such, findings may further serve to advance the scientific development of the entire field.

As stated in the introduction, there is a need to investigate interventions for veteran trauma survivors that may enhance engagement and/or outcomes associated with conventional treatments [17,18]. Further, these approaches might be shown through rigorous research to be equally or more effective compared to the treatments currently available. This need exists because treatment engagement is a significant challenge for conventional interventions [4,10,11,12] and as many as one-half of the veterans who do engage do not experience significant improvement [9]. However, for EAP interventions to meet this need, investigations of structured interventions that facilitate both fidelity to the model in replication studies and multi-site randomized controlled trails will be needed [13]. As a well-established model of EAP that utilizes a structured therapeutic framework and has a training and certification program, including military/veteran specific certification, the Eagala model [53,54,55,56] is a particularly good candidate to meet this need. Further, previous studies on community [65,66,67] and veteran [24,38] populations are promising. This investigation further assesses the suitability of this model for veteran trauma survivors and lays the foundation for future multi-site randomized controlled trials.

The first key finding of this study was that the model is feasible to implement and can potentially reach the target population across multiple sites. Over the one-year timeframe of data collection, services were provided for 107 veterans across 12 separate therapeutic equine facilities in the US. Further, 99% of the participants had experienced lifetime trauma, 87% had a PTSD diagnosis, 72% had been deployed, 69% had more than one mental health diagnosis, and 54% had experienced combat (Table 1). First, these feasibility results suggest that the intervention would be appropriate for future multi-site randomized controlled trails and, if shown to be beneficial, then could be successfully disseminated to the EAP field for veterans as an evidence-based intervention that utilizes a structured therapeutic framework. Second, the intervention appears to be able to reach the target population of veterans with trauma histories and/or psychiatric illness who may not engage with and/or respond to conventional treatments.

The second set of key findings from this study were related to treatment engagement. Given that challenges to treatment engagement are a major barrier associated with convention mental health treatments for this population [4,10,11,12], EAP interventions aimed to address this challenge would need to demonstrate treatment engagement better than that associated with conventional approaches. In this study, the average number of sessions attended was 5.96 out of six individual (98%) or 7.5 out of eight group (98%) sessions possible. Therapy completers are often defined as those attending ≥50% of the total sessions [68]; thus, most participants in this study were completers. In contrast, a large study of evidence-based psychotherapies for veterans with PTSD in Veterans Health Administration over a 15-year period [12] reported a completion rate of only 9.1%. Another measure of treatment engagement is client satisfaction. Mean CSQ-8 scores were 30.38 for individual and 29.20 for group sessions. Scores of 26–32 are interpreted to indicate high satisfaction [62]; thus, the participants were very satisfied with the intervention. Lastly, the CSQ-8 open-ended questions also supported client satisfaction as most respondents could not identify any program components that were unhelpful (Table 6). These results are consistent with other studies on EASs for veterans [34,35] and other populations [69] that suggest these programs are generally experienced as positive, which would likely support treatment engagement. While further research is needed, taken together, this and other studies suggest that EAS interventions in general, and the Eagala model of EAP in particular, might provide a solution to the problem of low conventional treatment engagement among veteran trauma survivors.

While rigorous research is needed to replicate the findings reported herein, the pre- to post-intervention improvements in psychological measures found in this investigation were very promising. Regarding symptom reduction, results revealed significant decreases in PTSD and depressive symptoms with moderate-to-large effect sizes (Table 2, Table 3 and Table 4). These findings are consistent with studies on various EAS interventions in both community [47,48,50] and veteran populations. Similar to the findings reported herein, veteran studies on EAS interventions have reported the benefits of both improved PTSD [14,23,24,25,27,29,30,31,32,36,37,38,40,41] and mood [23,27,28,30,33,34,35,36,37,38,40,43,44] symptoms. Additionally, this study found significant improvements in the trans-diagnostic metrics of functional impairment and satisfaction with life (Table 2, Table 3 and Table 4). This finding is also consistent with some other results [52,70,71,72] that suggest EAS interventions may have benefits beyond symptom reduction. Future studies will need to both replicate these findings as well as disambiguate the relationships between symptom reduction and trans-diagnostic outcomes.

Several findings of this investigation support not only the further development and evaluation of the Eagala model but also inform the entire field of EAP interventions for veterans. One unanswered question in the field of EAP for veterans is whether there are differences between group or individual treatment modalities regarding outcomes, cost-effectiveness, or participant preference. This study found no between-modality differences in PTSD symptom reduction or treatment satisfaction; however, individual therapy was associated with greater self-identified reduction in symptoms of depression and disability, as well as with improved satisfaction with life, even though participants received a lower dose of six versus eight sessions (Table 5). The importance of group interactions was a significant theme in response to open-ended questions (Table 6). In comparison, one other study [35] on a different EAS intervention reported on group versus individual therapy and found no differences in outcomes between these modalities. No firm conclusions can be drawn from these studies; however, the results suggest that there may be important differences between modalities in terms of both outcomes and treatment engagement, at least for some interventions. Future research will need to investigate these potential differences with a goal of developing guidelines for referral. Structured interventions, such as the Eagala model, which can be provided as either group or individual therapy, are ideally suited for these studies.

Lastly, an important issue for the entire AAI field is understanding the potential benefits and challenges associated with utilizing various species for this work. So, an important question for the EAP field to answer is, “why horses?” To our knowledge, no AAI investigations have compared horses to other therapy animals such as canines. Such studies will be needed to guide the use of various species in AAI in terms of outcomes, costs, and participant preferences. While this study did not directly address these issues, the greatest response to the CSQ open-ended questions under all three categories (Table 6) was “interaction with the horse.” This was under the theme of “most helpful” and supported by the prominent request for, “more time with the horses,” under the “suggestions for improvement” theme (Table 6). Thus, clearly the experience of the veteran participants was that the horse–human interactions were key to the benefits received. While studies comparing costs, safety, outcomes, and participant preferences for AAIs are needed, the results reported herein suggest that the Eagala model, and perhaps other EAP interventions, warrant further study as the AAI field advances.

This study had several limitations that must be considered when interpreting the results reported herein. Participants were self-referred, and it was an uncontrolled study; therefore, selection bias is a concern. Given the lack of both randomization and a control condition, cause-and-effect relationships were not established. The participants were all veterans or active-duty military personnel and predominately male and white. Thus, the results may not be generalized to other populations including civilian and other veteran or military populations. Further, test re-test bias cannot be ruled out for the results of the psychological instruments and the qualitative analysis did not systematically explore individual differences or group dynamics. Also, information about current mental health treatment and prior equine-assisted intervention participation was not collected. In this pilot study, it was not feasible to collect data regarding the specific activities that occurred during each session; thus, conclusions cannot be drawn regarding the relationships between specific activities and outcomes. Lastly, demographic and diagnostic information was self-reported and unverified. Nonetheless, the primary aim of this study to evaluate feasibility, treatment engagement, preliminary outcomes, and other characteristics associated with utilizing the Eagala model of EAP for military members or veterans with trauma histories was accomplished. The strength of the study was that it was a multi-site investigation with the largest sample size reported to date in the field of EAP for veterans.

## 5. Conclusions

The findings reported herein indicate that the Eagala model of EAP can be successfully implemented for active-duty military and veteran trauma survivors across multiple sites in the US. Further, treatment engagement may be better than that associated with conventional psychotherapy interventions for this population. Lastly, preliminary findings indicate that improvements in PTSD and depressive symptoms; additionally, decreased disability and enhanced satisfaction with life may be associated with participation. Future randomized controlled trails of this intervention are warranted and should include independent data verification and more diverse samples.

## Figures and Tables

**Table 1 ijerph-22-01557-t001:** Demographic and diagnostic characteristics of participants (*n* = 107).

Characteristic	*n*/M	%/SD
Age	46.23	12.63
Gender		
Male	70	65.4
Female	35	32.7
Race/Ethnicity		
White	66	61.7
Black/African Americans	12	11.2
Asian/Pacific Islander	4	3.7
Native American	2	1.9
Hispanic/Latino	12	11.2
Other	9	8.4
Branch		
Army	57	53.3
Marines	12	12.1
Navy	19	17.8
Air Force	16	15
Coast Guard	1	0.9
Deployed outside of US	77	72
Average number of deployments	1.75	1.85
Average number of OIF/OEF/OND deployments	0.88	1.16
Experienced combat	58	54.2
Experienced a trauma in life	106	99.1
Experienced combat trauma	81	75.7
PTSD Diagnosis	93	86.9
Additional MH diagnosis	74	69.2
Anxiety diagnosis	41	38.3
Depression diagnosis	57	53.3
Bipolar diagnosis	9	8.4

**Table 2 ijerph-22-01557-t002:** Pre- and post-intervention outcomes for the entire sample (*n* = 107).

	Pre-Test	Post-Test		95% Confidence Intervals		
Measure	M (SD)	M (SD)	*t*-Test	LL	UL	*d*
PCL-5	46.14 (17.38)	34.56 (18.42)	*t*(102) =7.22, *p* < 0.001	8.39	14.75	0.71
PHQ 9	14.14 (6.56)	10.01 (6.25)	*t*(103) = 7.29, *p* < 0.001	3.01	5.26	0.72
SWLS	15.56 (6.72)	18.85 (7.25)	*t*(100) = −2.14, *p* < 0.001	−4.44	−2.14	−0.57
SDS	17.70 (8.02)	14.01 (9.22)	*t*(101) = 4.911, *p* < 0.001	2.2	5.18	0.49

Note: LL = lower-level 95% confidence interval, UL = upper-level 95% confidence interval, and *d* = Cohen’s *d* effect size estimator.

**Table 3 ijerph-22-01557-t003:** Pre- and post-intervention outcomes for individual therapy (*n* = 52).

	Pre-Test	Post-Test		95% Confidence Intervals		
Measure	M (SD)	M (SD)	*t*-Test	LL	UL	*d*
PCL-5	46.65 (17.89)	31.63 (17.76)	*t*(48) = 6.11, *p* < 0.001	10.08	19.96	0.87
PHQ 9	14.44 (6.52)	8.64 (5.61)	*t*(49) = 6.20, *p* < 0.001	3.92	7.68	0.88
SWLS	15.98 (7.27)	20.39 (7.75)	*t*(48) = −5.32, *p* < 0.001	−6.08	−2.74	−0.76
SDS	16.60 (8.31)	11.60 (8.84)	*t*(49) = 3.93, *p* < 0.001	2.44	7.56	0.56

Note: LL = lower-level 95% confidence interval, UL = upper-level 95% confidence interval, and *d* = Cohen’s *d* effect size estimator.

**Table 4 ijerph-22-01557-t004:** Pre- and post-intervention outcomes for group therapy (*n* = 54).

	Pre-Test	Post-Test		95% Confidence Intervals		
Measure	M (SD)	M (SD)	*t*-Test	LL	UL	*d*
PCL-5	45.43 (17.20)	36.98 (18.86)	*t*(52) = 4.11, *p* < 0.001	4.32	12.58	0.56
PHQ 9	13.75 (6.66)	11.19 (6.63)	*t*(52) = 4.22, *p* < 0.001	1.35	3.79	0.58
SWLS	15.33 (6.15)	17.61 (6.37)	*t*(50) = −2.84, *p* = *0*.007	−3.88	−0.67	−0.4
SDS	18.57 (7.62)	16.20 (9.11)	*t*(50) = 2.94, *p* = 0.005	0.75	4	0.41

Note: LL = lower-level 95% confidence interval, UL = upper-level 95% confidence interval, and *d* = Cohen’s *d* effect size estimator.

**Table 5 ijerph-22-01557-t005:** Post-intervention outcomes by treatment modality (*n* = 106).

	Modality					
	Individual (*n* = 52)	Group (*n* = 54)		95% Confidence Intervals		
Measure	M (SD)	M (SD)	*t*-Test	LL	UL	*d*
CSQ-8	30.38 (2.35)	29.20 (4.38)	*t*(104) = 1.72, *p* = 0.11	0.01	2.44	0.33
SDS	11.75 (8.81)	16.06 (8.95)	*t*(103) = −2.49, *p* = 0.02	−7.75	−0.87	−0.49
SWL	20.90 (7.91)	17.81 (6.33)	*t*(103) = 2.21, *p* = 0.03	0.32	5.86	0.43
PHQ 9	8.52 (5.57)	11.11 (6.60)	*t*(104) = −2.18, *p* = 03	−4.95	−0.24	−0.42
PCL-5	31.29 (17.60)	37.17 (18.73)	*t*(103) = −1.65, *p* = 0.10	−12.92	1.17	−0.32

Note: LL = lower-level 95% confidence interval, UL = upper-level 95% confidence interval, and *d* = Cohen’s *d* effect size estimator. At pre-intervention, there were no differences in these measures by group.

**Table 6 ijerph-22-01557-t006:** Responses to CSQ-8 open-ended survey questions (*n* = 107).

What Parts of the Program Were Most Helpful to You?	What Parts of the Program Were Least Helpful to You?	What Suggestions Do You Have for How the Program Can be Improved?
Interaction with the horses (*n* = 64)	None/Nothing (*n* = 33)	None/Nothing (*n* = 29)
Processing emotions and thoughts (*n* = 13)	Did not answer (*n* =16)	More and/or more frequent sessions (*n* = 25)
Group interactions (*n* = 12)	Not enough sessions (*n* = 11)	Different activities (*n* = 13)
Therapist interaction (*n* = 8)		More time with horses (*n* = 9)
Relationship building and trust (*n* = 7)		Increased expectation setting/Instruction (*n* = 8)
Symbolism/metaphors (*n* = 6)		Did not answer/DNA (*n* = 7)
Feeling mindful and/or present (*n* = 3)		Program promotion (*n* = 3)
Insight/Awareness (*n* = 3)		More veteran-to-veteran interactions (*n* = 2)

Note. Participant responses may have been coded into multiple themes; as such, the number of themes identified in each column may exceed the total number of participants.

**Table 7 ijerph-22-01557-t007:** Qualitative themes and subthemes of CSQ-8 open-ended questions.

Themes	Subthemes
Most Helpful Parts of the Program	Interaction with horses
	Symbolism/metaphors
	Feeling mindful and/or present
	Therapist interaction
	Processing emotions/thoughts
	Relationship building/trust
	Insight/Awareness
	Group interactions
Least Helpful Parts of the Program	Challenges with some of the activities
	Not enough time or sessions
	None/Nothing
	Did not answer/DNA
Suggestions for program improvement	Increased expectation setting/instruction
	More and/or more frequent sessions
	Different activities
	More veteran-to-veteran interactions
	Program promotion
	Improve accessibility and change in consistency
	None/Nothing
	Did not answer/DNA

## Data Availability

The dataset is available on request from the authors.

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
