# Peer review of "A Multi-Site Observational Evaluation of the Equine Assisted Growth and Learning Association Model of Equine-Assisted Psychotherapy for Veteran Trauma Survivors"

_ijerph, 2025, doi:10.3390/ijerph22101557_

Round 1

Reviewer 1 Report

Comments and Suggestions for Authors

Please see attached pdf.

Author Response

Overarching thoughts and comments:

- Conceptually strong and frankly, the most well-written paper I have reviewed in at least 2 years – thank you.
-Thank you for your kind comment!
- Please review (especially the intro and methods) for consistent tense usage. Currently language oscillates from past to present, and back.
-We have reviewed and did not find incorrect use of past and present tense. If it is felt that this still needs to be addressed further, we would be willing to do so in a subsequent revision.

- In my field or practice in mental health and EAS, I am careful to be very specific when I am referring to “post-traumatic stress” vs. PTSD. PTSD is a diagnosis, and a controversial one at that: Consider Smith and Whooley (2015) who advocate to “drop the D”, and the movement to focus on post-traumatic growth rather. Moreover, moral injury is often a focus of post-war stress and trauma rather than PTSD. If this study is specifically researching a participant population that has a PTSD diagnosis (inclusion criteria) and changes overtime based on the intervention (EFP), then PTSD is accurate and acceptable. If the outcome is based on self-identified symptom reduction, with no reference to diagnosis then the use of PTSD is irrelevant and may be potentially harmful. I recommend that the authors consider their participant population and include a rationale (in the introduction) for the language used.
- Consider: Smith, R. T., & Whooley, O. (2015). Dropping the disorder in PTSD. Contexts, 14(4), 38-43.
-If you choose to use PTSD, you may want to reference that this is relevant based on the population served (e.g. 87 % had a self-reported PTSD diagnosis.

-We agree with the challenges of terminology regarding the use of “PTSD.” We have clarified the introduction section to indicate that the study is focused on veteran trauma survivors, some of which have a diagnosis of PTSD. Also, please see our
response to a comment below. We believe this change addresses this comment as well. If it is felt that this still needs to be addressed further, we would be willing to do so in a subsequent revision.

- If you have information about whether participants were receiving other mental health supports (via the VA or community), please include.
-We do not have this information. We have added mention of this in the limitations paragraph of the discussion section. Line 485- 486.

- If you have information about whether this intervention was the first equine interaction for participants, please include.
-We do not have this information. We have added mention of this in the limitations paragraph of the discussion section. Line 485- 486.

- The explanation of Eagala is not sufficient as currently written. Based on current content, you could just as easily be discussing ESMHL model of PATH, Natural Lifemanship ground-work, or HERD Institute EFP sessions. Please write at least one paragraph in the introduction explaining the model in detail (beyond use of metaphors).
-We have expanded the explanation of the Eagala model. In response to another comment, this information is now located in the introduction section. Line 96 - 178.

As a mental health practitioner of EAS and a researcher of EAS, I appreciate that you detail the EAGALA model (MH + ES) and define modalities (EAS, EAL, EFP). It is important to focus the terminology intended for the inter-vention. You don't have to be "all inclusive", I would rather you be specific; if this is an EFP intervention be spe-cific about the language and the intervention throughout. Similarly, if the intervention is manualized, the specific of the sessions needs to be shared (or cited to a website to learn more about the manual) so that the intervention can be effectively replicated in future efforts. The EAS field is mixed on the need, use, and sustainability of manu-als for EFP (or other) activities/interventions. Thus, this requires specific language to explain the need, purpose, and intention of using a manual.
- We agree and have made changes to the use of terminology throughout the manuscript in response to this and other comments.

Specific comments line-by-line:

- Line 61 and Line 73: Is EAS for military persons/PTSD still considered novel? What does it mean for the field to be in “early stages of scientific development”? Many, myself included, would argue that is it under-resourced thus underdeveloped - but not “new”. I recommend making this clear to differentiate between new and understudied.
-We have deleted the term “novel” from the text.
-We have changed “early stages of scientific development” to “under-resourced and thus underdeveloped.” Line 75 - 76.

- Line 69: make “utilized” present tense for consistency with other content
-We have made this change.

- Line 80: the terms “previously reviewed” has already been used 3 times, I suggest varying your language for flow
-We have made this change.

- Line 83, 398: manualization: needs to be defined. I highly recommend adding your argument for why manualization is good for the field, not just research. Many practitioners do not like the idea of a manual as they want to: meet the client where they are at, flex based on the horse and human needs, allow for creativity in the moment… A manual like Hoagwood et al (2022) or Gabriels et al (2015) accounts for every minute in a session and what the EAS (and MH) practitioners must. Is that what you are suggesting is needed? As a researcher, I agree that there is a need for manualization to insure fidelity and rigor in research efforts. However, as a practitioner, I do not agree that the field should or could function is all experiences are required to follow a manual.
-Hoagwood, K., Vincent, A., Acri, M., Morrissey, M., Seibel, L., Guo, F., ... & Horwitz, S. (2022). Reducing anxiety and stress among youth in a CBT-based equine-assisted adaptive riding program. Animals, 12(19), 2491.
-Gabriels, R. L., Pan, Z., Dechant, B., Agnew, J. A., Brim, N., & Mesibov, G. (2015). Randomized controlled trial of therapeutic horseback riding in children and adolescents with autism spectrum disorder. Journal of the American Academy
of Child & Adolescent Psychiatry, 54(7), 541-549.
-Since we are no longer using the term manualized in this revision (see response below), we did not address this comment. If it is felt that this still needs to be addressed, we would be willing to do so in a subsequent revision.

- Line 104: Continuing my comment above, Eagala in and of itself is not a manualized intervention. It is a training and credentialing program that provides practitioners with framework to practice (e.g. metaphors) and skill (e.g. vocabulary and facilitation) to co-facilitate (EAS, MH, and horse(s)) sessions with a range of activities. Practitioners do not have to practice
using specific activities in a particular order or structure. It is misleading to call Eagala a manualized intervention.
-We agree and have changed “manualized” to “structured” or “structured therapeutic framework” throughout the manuscript.

- Line 125: The term metaphor has been used three times but not het described nor explained. To someone outside of the EAS industry, this needs explanation.
-This now defined in the introduction section, lines 153 and 154.

- Line 128: Emotions and cognitions are not tangible. Please use a different word. Do you mean visceral?
-This was changed with the rewritten introduction section explaining the Eagala model. Line 96 - 178.

- Line 131: This is important as to “why horses?”. Horses are more than non-judgmental. Horses are co-facilitator in Eagala and warrant more than one sentence as to their role, purpose, and potential impact.
-We have added significantly to the discussion of “why horses” in the explanation of the Eagala model in the introduction section. Line 104 -115.

- Lines 102-132: Consider moving into the introduction so the Methods section is strictly focused on the methods
-We agree and have made this change.

- Line 144: when discussing consent, include IRB approval #
-We made this change. Line 210. Also, The IRB number and additional approval information is given in the “Institutional Review Board Statement” located after the main text.

- Line 147: Why wouldn’t you have included a diagnosis of PTSD as the inclusion criteria since that is your population on inquiry. “The only inclusion criteria for data collection were active duty or military veteran status and having a history of trauma and self-identify as having a disability.”
-To that end, I recommend running your statistical tests only with those who have a self- reported PTSD (identified as 87%). The inclusion of those who do not have PTSD is counterintuitive to your framing (introduction) and intent.
-A diagnosis of PTSD was not an inclusion criterion as the clinical intervention provided was funded (not funding for this research study) by a grant that specified the inclusion criteria and did not allow limiting the participants to those with a PTSD
diagnosis.
-We agree there was a mismatch between the framing in the introduction and the population studied. Thus, we have made edits to the first two paragraphs of the introduction to clarify the study was about veteran trauma survivors in general and not just those with a diagnosis of PTSD. Having made that change, we did not rerun the statistics. If it is felt that it still needs to be addressed, we would be willing to do so in a subsequent revision.

- Line 158: Who administered the PTSD checklist? Do you have any concern (limitations, perhaps) about test re-test bias as this is a very common toll used in the VA and other mental health settings.
- The PCL and all instruments were administered by the mental health clinician facilitating the intervention. Line 214 - 215. This information has been added to the methods section. We agree that test re-test bias is a possibility and have added this as a limitation.

- Line 162: When stating PCL-5, were you using the Military, Civilian, or Specific measure: https://www.ptsd.va.gov/professional/assessment/adult-sr/ptsd-checklist.asp
-The Standard Civilian version was used. This has been clarified in the methods section. Line 215 & 219.

- Line 201: “a person”, a statistician? Member of the research team?
-This has been changed to “four members of the research team.” Line 254.

- Line 208: While duration is shared, there is no explanation of what occurred in the session. This is particularly important since you are asserting that Eagala is manualized. As such, there should be a table with a high level overview of the activities in each of the 6 sessions, consistent at all sites. It is understood that the actual Eagala training is proprietary, but as research, there needs to be information to consider replication. Stating: 1. Grooming at liberty; 2. Obstacle course; 3. Constellations (explain); 4. Trail walk… is sufficient. As a final point on this matter, the horse is missing throughout your writing. Yes, you address “why horses?” in your discussion, but they are absent from much of your argument. Adding this content will bring the horse, as an important cofacilitator into the framework.
-We have changed the terminology in response to another comment and are no longer using the term “manualized.”
-We have added to the discussion about “why horses” in the introduction. Line 104 - 115.
-Regarding what happened in sessions, we have explained in the introduction and methods section that the Eagala model does not specify session activities and that for this pilot study it was not feasible to collect session activity data. We have also added the lack of information about specific session activities to the limitations section. We believe the expanded description of the Eagala model in the introduction provides a clear picture for readers of possible session activities.

- Line 209: what was the max number of horses in a session?
-We did not collect this information. We have indicated this in the first paragraph of the results section.

- Line 242 & 389: ”greater reductions in disability” – I think this refers to self-identified symptom reduction? Disabilities do not necessarily reduce, nor is the Eagala intervention intended to reduce the disability – it is intended to reduce the symptoms experienced by the participant that may be related to the disability.
-We have changed the wording as recommended.

- Line 246-248: please include N after referring to the sub groups
-We have made this change.

- Table 1: Please use left align for text
- We have made this change.

- Table 7: I recommend writing as a paragraph and not using this table
-We appreciate the reviewer’s thoughts but feel the table is more reader friendly. If it is felt that it still needs to be addressed, we would be willing to do so in a subsequent revision.

- Line 367: I recommend changing EAS to EFP (here and throughout the paper when referring to the Eagala model as EAS). The sessions are mental health focus, with a licensed MH practitioner co-facilitating, with the intent of decreasing symptoms if a MH diagnosis….that is EFP.
-We use equine-assisted psychotherapy (EAP) in the original manuscript. We have replaced EAS with EAP except in situations where we are referring to the broader field.

- Line 400: True, the field is primed for a comparative study. Consider using prior AAI research to strengthen your argument: 1. Payne, E., DeAraugo, J., Bennett, P., & McGreevy, P. (2016). Exploring the existence and potential underpinnings of dog–human and horse–human attachment bonds. Behavioural processes, 125, 114-121. https://www.sciencedirect.com/science/article/abs/pii/S0376635715300498;

2. Animal compared to toy: O'Haire, M. E., McKenzie, S. J., Beck, A. M., & Slaughter, V. (2013). Social behaviors increase in children with autism in the presence of animals compared to toys. PloS one, 8(2), e57010.;

3. Guinea pig vs waitlist O'Haire, M. E., McKenzie, S. J., McCune, S., & Slaughter, V. (2013). Effects of animal-assisted activities with guinea pigs in the primary school classroom. Anthrozoös, 26(3), 445-458.

-We agree that reviewing AAI research could strengthen our argument. However, we feel additional review is beyond the scope of this already lengthy paper. If it is felt that it still needs to be addressed, we would be willing to do so in a subsequent revision.

Reviewer 2 Report

Comments and Suggestions for Authors

General comment: The manuscript “A Multi-site Observational Evaluation of the Equine Assisted  Growth and Learning Association Model of Equine-assisted  Psychotherapy for Veteran Trauma Survivorsis an interesting paper reporting a retrospective multi-site observational study  whose  main purpose was to evaluate the feasibility, safety, and preliminary outcomes of the Equine Assisted Growth and Learning Association (Eagala) model of equine-assisted psychotherapy for active-duty military personnel and veterans with trauma histories. The Eagala model of equine-assisted therapy is a manualized, experiential, and solution-focused therapeutic approach that incorporates horses to help clients address emotional, psychological, and behavioral challenges.  

The specific objectives of the study were to examine the implementation of the Eagala model in order to assess whether the model could be successfully applied across multiple sites in the United States and to evaluate participant adherence to the program and engagement levels; to evaluate the therapeutic outcomes, by measuring changes in PTSD symptoms, depression, functional disability, and life satisfaction before and after the intervention; to ​analyze the participant satisfaction with the program; and, particularly, to ​contribute to the scientific development of the field, by providing preliminary data to support future randomized controlled trials, and by​ exploring the potential of the Eagala model as a manualized and evidence-based intervention for veterans with trauma. ​ The study was carried out on 107 participants at 12 sites. Participants ranged in age from 22 to 78 and were predominately male.

The study used the following psychological instruments to evaluate participants pre- and post-intervention: a PTSD Checklist for DSM-5 (PCL-5), ​that measured the severity of PTSD symptoms; a Patient Health Questionnaire-9 (PHQ-9), that assessed the severity of depressive symptoms; a Satisfaction with Life Scale (SWLS), that measured life satisfaction; the Sheehan Disability Scale (SDS), that evaluated functional impairment in work/school, social life, and family life/home responsibilities; a Client Satisfaction Questionnaire-8 (CSQ-8), that measured participant satisfaction with the intervention.​ These instruments provided quantitative data on PTSD symptoms, depression, life satisfaction, functional impairment, and treatment satisfaction.

The methodology adopted for the research has strengths and some limitations that need to be considered: the research is a multi-site study that was conducted across 12 therapeutic equine facilities in the United States, ensuring greater geographic representation and participant diversity. ​This approach facilitates the evaluation of the feasibility of the Eagala model on a larger scale. ​With 107 participants, the study represents the largest sample size reported to date in the field of equine-assisted services for veterans, increasing the robustness of the findings. ​

Moreover, the study adopted the use of validated psychological instruments (PCL-5, PHQ-9, SWLS, SDS) to assess changes in PTSD symptoms, depression, disability, and life satisfaction, a ​quantitative analysis using t-tests to compare pre- and post-intervention results, supported by effect size measures (Cohen's d),a  â€‹qualitative analysis by the use of the RAPID method to analyze open-ended responses from the CSQ-8 questionnaire, providing deeper insights into participants' experiences, and a​ focus on satisfaction and engagement, by measuring the evaluation of participant satisfaction and session completion rates, highlighting the model's effectiveness in engaging veterans.

However, the study also showed some limitations, like the lack of a control group. The absence of randomization and a control group limits the ability to establish causal relationships between the intervention and observed outcomes. ​Moreover, the participants were self-selected, introducing potential selection bias. ​ This may affect the generalizability of the findings. ​The demographic and diagnostic information was self-reported by participants without independent verification, which could compromise data accuracy. T​he program was relatively short (6-8 sessions), and some participants indicated that the timeframe was insufficient to achieve full benefits. ​Most participants were white male veterans, limiting the generalizability of the findings to other populations, such as female veterans or civilians with trauma. ​

Finally, while useful, the qualitative analysis did not systematically explore individual differences or group dynamics. In conclusion, the methodology adopted is suitable for a preliminary study aimed at assessing the feasibility and initial outcomes of the Eagala model. ​ However, to strengthen the validity of the findings, future studies with more rigorous designs, such as randomized controlled trials, independent data verification, and more diverse samples, are necessary. ​ In summary, the study aims to demonstrate that the Eagala model is a feasible and promising therapy for improving the psychological well-being and quality of life of veterans, with the goal of encouraging further research and applications in the field. In fact, the Eagala model stands out for its use of horses, metaphorical and experiential techniques, team-based structure, and focus on emotional processing in a non-judgmental, ground-based setting. ​ These elements make it a unique and engaging alternative to conventional therapies.

 From my point of view, the paper is interesting, because the contents addressed in this study are worthy of further investigations, from both the speculative and the applied points of view. The conclusions drawn, in fact, suggest using the in future horse-human research to determine if the outcomes are consistent. This could be a valuable tool in evaluating the effects of human-animal relationships in order to improve humans’ welfare. Moreover, the Authors warrant future randomized controlled trails of this intervention.

Title: It is suitable and it well describes the experiment presented in the manuscript.

Abstract: It is suitable. Abstract clearly identifies the interest for this research and its possible relevance. It recaps the information contained in the main text without repetitions.

Introduction: The Introduction provides adequate background. This section is concise, and, although limited, it includes some specific literature references. However, further literature references could be added.

Materials and Methods: The main concern of the paper regards this section. In fact, as reported in the comments, the methodology adopted for the research has strengths and some limitations that need to be considered. The main limitation of this work is the lack of a control group. The absence of randomization and a control group limits the ability to establish causal relationships between the intervention and observed outcomes. ​Moreover, the participants were self-selected, introducing potential selection bias. ​ This may affect the generalizability of the findings. ​The demographic and diagnostic information was self-reported by participants without independent verification, which could compromise data accuracy. T​he program was relatively short (6-8 sessions), and some participants indicated that the timeframe was insufficient to achieve full benefits. ​Most participants were white male veterans, limiting the generalizability of the findings to other populations, such as female veterans or civilians with trauma. ​Finally, while useful, the qualitative analysis did not systematically explore individual differences or group dynamics. However, the study also showed some limitations, like the lack of a control group. The absence of randomization and a control group limits the ability to establish causal relationships between the intervention and observed outcomes. ​Moreover, the participants were self-selected, introducing potential selection bias. ​ This may affect the generalizability of the findings. ​The demographic and diagnostic information was self-reported by participants without independent verification, which could compromise data accuracy. T​he program was relatively short (6-8 sessions), and some participants indicated that the timeframe was insufficient to achieve full benefits. ​Most participants were white male veterans, limiting the generalizability of the findings to other populations, such as female veterans or civilians with trauma. ​Finally, while useful, the qualitative analysis did not systematically explore individual differences or group dynamics. In conclusion, the methodology adopted is suitable for a preliminary study aimed at assessing the feasibility and initial outcomes of the Eagala model. ​ However, to strengthen the validity of the findings, future studies with more rigorous designs, such as randomized controlled trials, independent data verification, and more diverse samples, are necessary.

Results: The results are clear. Results are presented clearly and logically, and the data are clearly presented in the figures and tables.

Discussion: The discussion is well organized and balanced. The comments reported in discussion are pertinent to the results achieved. The authors critically examine their results in the light of the state of science highlighted in the introduction. The discussion of results is extensive and clear.  Discussion follows a logical line, but the current knowledge on the topic could be more properly presented studying and adding further references about from the horses’ point of view. The conclusions are drawn from the study related to the aim of the study and potentially plausible in terms of the results obtained and applied in equine assisted services.  The interpretation of results proposed by the authors in the discussion are interesting. The paper offers the perspective for further study.

References: They are appropriate, although they could be reinforced by further items.

Tables and Figures: They are clear and explicative.

Decision: The current manuscript, although the methodology adopted for the research has strengths and some limitations that need to be considered, could be acceptable for publication after minor revision.

Author Response

General comment:

The manuscript "A Multi-site Observational Evaluation of the Equine Assisted Growth and Learning Association Model of Equine-assisted Psychotherapy for Veteran Trauma Survivors" is an interesting paper reporting a retrospective multi-site observational study whose main purpose was to evaluate the feasibility, safety, and preliminary outcomes of the Equine Assisted Growth and Learning Association (Eagala) model of equine-assisted psychotherapy for active-duty military personnel and veterans with trauma histories. The Eagala model of equine-assisted therapy is a manualized, experiential, and solution­ focused therapeutic approach that incorporates horses to help clients address emotional, psychological, and behavioral challenges.

The specific objectives of the study were to examine the implementation of the Eagala model in order to assess whether the model could be successfully applied across multiple sites in the United States and to evaluate participant adherence to the program and engagement levels; to evaluate the therapeutic outcomes, by measuring changes in PTSD symptoms, depression, functional disability, and life satisfaction before and after the intervention; to analyze the participant satisfaction with the program; and, particularly, to contribute to the scientific development of the field, by providing preliminary data to support future randomized controlled trials, and by exploring the potential of the Eagala model as a manualized and evidence-based intervention for veterans with trauma. The study was carried out on 107 participants at 12 sites. Participants ranged in age from 22 to 78 and were predominately male.

The study used the following psychological instruments to evaluate participants pre- and post-intervention: a PTSD Checklist for DSM- 5 (PCL-5), that measured the severity of PTSD symptoms; a Patient Health Questionnaire-9 (PHQ-9), that assessed the severity of depressive symptoms; a Satisfaction with Life Scale (SWLS), that measured life satisfaction; the Sheehan Disability Scale (SOS), that evaluated functional impairment in work/school, social life, and family life/home responsibilities; a Client Satisfaction Questionnaire-8 (CSQ-8), that measured participant satisfaction with the intervention. These instruments provided quantitative data on PTSD symptoms, depression, life satisfaction, functional impairment, and treatment satisfaction.

The methodology adopted for the research has strengths and some limitations that need to be considered: the research is a multi-site study that was conducted across 12 therapeutic equine facilities in the United States, ensuring greater geographic representation and participant diversity. This approach facilitates the evaluation of the feasibility of the Eagala model on a larger scale. With 107 participants, the study represents the largest sample size reported to date in the field of equine-assisted services for veterans, increasing the robustness of the findings.

Moreover, the study adopted the use of validated psychological instruments (PCL-5, PHQ-9, SWLS, SOS) to assess changes in PTSD symptoms, depression, disability, and life satisfaction, a quantitative analysis using t-tests to compare pre- and post­ intervention results, supported by effect size measures (Cohen's d),a qualitative analysis by the use of the RAPID method to analyze open-ended responses from the CSQ-8 questionnaire, providing deeper insights into participants' experiences, and a focus on satisfaction and engagement, by measuring the evaluation of participant satisfaction and session completion rates, highlighting the model's effectiveness in engaging veterans.

However, the study also showed some limitations, like the lack of a control group. The absence of randomization and a control group limits the ability to establish causal relationships between the intervention and observed outcomes. Moreover, the participants were self-selected, introducing potential selection bias. This may affect the generalizability of the findings. The demographic and diagnostic information was self-reported by participants without independent verification, which could compromise data accuracy. The program was relatively short (6-8 sessions), and some participants indicated that the timeframe was insufficient to achieve full benefits. Most participants were white male veterans, limiting the generalizability of the findings to other populations, such as female veterans or civilians with trauma.

Finally, while useful, the qualitative analysis did not systematically explore individual differences or group dynamics. In conclusion, the methodology adopted is suitable for a preliminary study aimed at assessing the feasibility and initial outcomes of the Eagala model.

However, to strengthen the validity of the findings, future studies with more rigorous designs, such as randomized controlled trials, independent data verification, and more diverse samples, are necessary. In summary, the study aims to demonstrate that the Eagala model is a feasible and promising therapy for improving the psychological well-being and quality of life of veterans, with the goal of encouraging further research and applications in the field. In fact, the Eagala model stands out for its use of horses, metaphorical and experiential techniques, team-based structure, and focus on emotional processing in a non-judgmental, ground­ based setting. These elements make it a unique and engaging alternative to conventional therapies.

From my point of view, the paper is interesting, because the contents addressed in this study are worthy of further investigations, from both the speculative and the applied points of view. The conclusions drawn, in fact, suggest using the in future horse-human research to determine if the outcomes are consistent. This could be a valuable tool in evaluating the effects of human­ animal relationships in order to improve humans' welfare.  Moreover, the Authors warrant future randomized controlled trails of this intervention.

            -Thank you so much for your very thorough review of our manuscript.

- Title: It is suitable and it well describes the experiment presented in the manuscript.

            -Thank you!

- Abstract: It is suitable. Abstract clearly identifies the interest for this research and its possible relevance. It recaps the information contained in the main text without repetitions.

            -Thank you!

Introduction: The Introduction provides adequate background. This section is concise, and, although limited, it includes some specific literature references. However, further literature references could be added.

            -Thank you!

- Materials and Methods: The main concern of the paper regards this section. In fact, as reported in the comments, the methodology adopted for the research has strengths and some limitations that need to be considered. The main limitation of this work is the lack of a control group. The absence of randomization and a control group limits the ability to establish causal relationships between the intervention and observed outcomes. Moreover, the participants were self-selected, introducing potential selection bias. This may affect the generalizability of the findings. The demographic and diagnostic information was self-reported by participants without independent verification, which could compromise data accuracy. The program was relatively short (6-8 sessions), and some participants indicated that the timeframe was insufficient to achieve full benefits. Most participants were white male veterans, limiting the generalizability of the findings to other populations, such as female veterans or civilians with trauma. Finally, while useful, the qualitative analysis did not systematically explore individual differences or group dynamics. However, the study also showed some limitations, like the lack of a control group. The absence of randomization and a control group limits the ability to establish causal relationships between the intervention and observed outcomes. Moreover, the participants were self-selected, introducing potential selection bias. This may affect the generalizability of the findings. The demographic and diagnostic information was self-reported by participants without independent verification, which could compromise data accuracy. The program was relatively short (6-8 sessions), and some participants indicated that the timeframe was insufficient to achieve full benefits. Most participants were white male veterans, limiting the generalizability of the findings to other populations, such as female veterans or civilians with trauma. Finally, while useful, the qualitative analysis did not systematically explore individual differences or group dynamics. In conclusion, the methodology adopted is suitable for a preliminary study aimed at assessing the feasibility and initial outcomes of the Eagala model. However, to strengthen the validity of the findings, future studies with more rigorous designs, such as randomized controlled trials, independent data verification, and more diverse samples, are necessary.

            -We agree regarding limitations of the study.  Many of these were stated in the original manuscript.  We have added limitations regarding generalizability to civilian populations and that the qualitative analysis did not systematically explore individual differences or group dynamics to the limitations paragraph of the discussion section.  We have also added that future studies should include independent data verification, and more diverse samples to the discussion section. 

Results: The results are clear. Results are presented clearly and logically, and the data are clearly presented in the figures and tables.

            -Thank you!

Discussion: The discussion is well organized and balanced. The comments reported in discussion are pertinent to the results achieved. The authors critically examine their results in the light of the state of science highlighted in the introduction. The discussion of results is extensive and clear. Discussion follows a logical line, but the current knowledge on the topic could be more properly presented studying and adding further references about from the horses' point of view. The conclusions are drawn from the study related to the aim of the study and potentially plausible in terms of the results obtained and applied in equine assisted services. The interpretation of results proposed by the authors in the discussion are interesting. The paper offers the perspective for further study.

            -Thank you!

References: They are appropriate, although they could be reinforced by further items.

           -We reviewed our references and did not think of any specific literature that should be added. If it is felt that it still needs to be addressed, we would be willing to do so in a subsequent revision.

Tables and Figures: They are clear and explicative.

            -Thank you!

Decision: The current manuscript, although the methodology adopted for the research has strengths and some limitations that need to be considered, could be acceptable for publication after minor revision.

            -Thank you!